# Research on the evolution mechanism of main control factors for coalbed methane extraction

**Ting Xia, Enyu Xu, Xijian Li** **\*, Shoukun Chen**

Mining College, Guizhou University, Guiyang, China

\* xjli1@gzu.edu.cn

## Abstract

To study the isothermal adsorption and desorption diffusion processes of anthracite, isothermal adsorption and desorption experiments of coal powder were conducted, and numerical simulations were carried out using COMSOL Multiphysics. The results showed that the DR equation had the best fitting effect on the gas adsorption curve, followed by the Langmuir equation, and the BET equation had the worst fitting effect, with $R^2$ values of 0.9986, 0.9976, and 0.9765, respectively. The potential reason for this is the development of micropores in anthracite, with a relatively large proportion of gas molecules are primarily adsorbed in micropores; The single-pore model is suitable for fitting the gas adsorption process, but not for fitting the gas desorption process. Taking into account the spatial and temporal evolution characteristics of the gas diffusion coefficient, the dual-pore model is more suitable for simulating gas migration at the coal seam scale; Gas adsorption can cause expansion and deformation of the coal matrix, and some molecular scale pore throats have smaller pore throats than gas molecules after adsorption and expansion, resulting in a blocking effect on gas molecules in the pores. Therefore, the amount of gas desorption is usually smaller than the adsorption amount; There are specific differences between gas adsorption and desorption processes. Gas adsorption is a constant volume condition, while desorption is a constant pressure condition. Therefore, adsorption and desorption cannot be simply regarded as reversible processes. The research results provide theoretical support for a deeper understanding of the processes and mechanisms of gas adsorption-diffusion and desorption-diffusion.

## 1 Introduction

Gas prevention and control was an essential task in coal mining. It was of great significance for both resource utilization and safe production [1–3]. Exploration studies indicated that China's coalbed methane reserves exceeded forty trillion cubic feet, showing great potential for use. However, the dual-porosity structure of coal media results in highly heterogeneous characteristics for gas storage and migration. The

**Data availability statement:** "All relevant data are within the paper.".

**Funding:** This work was supported by the National Nature Science Foundation of China (Grant No. 52164015, 52364009), Guizhou Province Science and Technology Achievement Transformation Joint Fund Project [Qian Ke He Cheng Guo LH (2025) Key 002], Guizhou Provincial Basic Research Program (Natural Science) (No. Qian Ke He Ji Chu - ZK [2024] Yi Ban 098), Guizhou Provincial Postgraduate Research Funding Program (2024YJSKYJJ082). The funders had no role in study design, data collection and analysis, decision to publish, or preparation of the manuscript. All the funding or sources of support (whether external or internal to your organization) received during this study. There was no additional external funding received for this study.

**Competing interests:** The authors have declared that no competing interests exist.

complexity can be attributed to three points: Firstly, the permeability difference of the micropore-fissure system reaches 3–4 orders of magnitude; Secondly, the dynamic contraction of pore diameters caused by the coupling of effective stress and adsorption expansion; Thirdly, the patchy distribution of in-situ gas-water two-phase flow in space. Particularly crucially, gas molecules are mainly bound in the < 2 nm matrix micropores, and their effective diffusion coefficient changes synchronously with the expansion-contraction effect, becoming the core bottleneck of the desorption-diffusion-permeation coupling process [4–6]. Recent studies have shown that the coal matrix has nano-pores with a large specific surface area and pore volume, providing conditions for the occurrence of gas molecules [7–9]. Therefore, the study of gas adsorption-diffusion and desorption-diffusion processes was of great importance.

Many scholars have respectively expounded the laws of coal body gas adsorption/desorption from perspectives such as coal rank, pore structure, fractal model, structural deformation, adsorption heat effect, and acidification modification [10]. Wang Qingqing et al [11] studied the relationship between coal rank and gas adsorption/desorption characteristics parameters based on experiments and field tests. Li Weibo et al [12] found the threshold entry pressure for gas in vitrain (0.20-1.03 MPa) and its median capillary pressure (8.16-10.14 MPa) are 3–15 and 1.5-2.4 times higher, respectively, than those in durain, indicating a significantly elevated barrier to gas invasion. Mercury-withdrawal efficiency is only 25.46-41.85%, lower than that of durain, demonstrating a high proportion of ink-bottle or semi-closed pores. The mercury-injection curve continues to rise at high pressures without reaching a plateau, and the nitrogen hysteresis loop is 15–25% larger than in durain; together, these features confirm that the pore throats are highly tortuous and the connectivity is restricted. Jin Yi et al [13] proposed a single – layer adsorption model based on the fractal topology theory, and pointed out the logarithmic, exponential and linear evolution laws of adsorption coverage. Based on isothermal adsorption experiments and pore structure tests, Meng Zhaoping et al [14] found that tectonic coal had better gas migration ability than primary coal, and divided the coal matrix gas diffusion process into three stages. Li Yaochen et al [15] investigated the effect of coal matrix adsorption heat on $CO_2$ geological storage. Results showed that when the gas injection pressure was constant, adsorption heat reduced the gas injection efficiency by 16.8%. Yuan Mei et al [16] found that acidification can enhance coal matrix pore structure and connectivity, thus increasing the coal matrix gas diffusion rate and reducing its adsorption capacity. Sun Xiaoxiao et al [17] found in the $CO_2$ displacement of coalbed $CH_4$ that both sweeping and replacement effects together determined the displacement efficiency, with the former mainly in the early stage and the latter in the later stage.

Numerous studies have predominantly concentrated on the gas adsorption capacity of coal matrix, yet neglected the specific adsorption process. Given the distinct differences between adsorption and desorption processes, prior research was deemed insufficient. This study carried out experiments on gas adsorption – diffusion and desorption – diffusion in granular coal and executed numerical simulations based on the experimental findings, delving deeply into the gas diffusion process in granular

coal. The research outcomes could furnish a theoretical basis for gaining an in – depth understanding of the occurrence and migration mechanisms of gas in coal matrix.

## 2 Experiment

### 2.1 Sample preparation

The fresh coal samples from the underground are encapsulated and then transported to the laboratory. They are then crushed and sieved to obtain 50g of coal powder with a particle size of 60–80 mesh for conducting gas adsorption and desorption experiments.

### 2.2 Scanning electron microscope test

The apparent morphology of the coal samples was observed using a scanning electron microscope, as shown in Fig 1. It can be seen that the surface of the coal samples is rough and is covered with various-sized and-shaped pore networks, providing a vast space for the adsorption of methane gas in the coal. At the same time, as a diffusion and mass transfer channel for $CH_4$ molecules, it causes the complex dynamic evolution characteristics of the diffusion process.

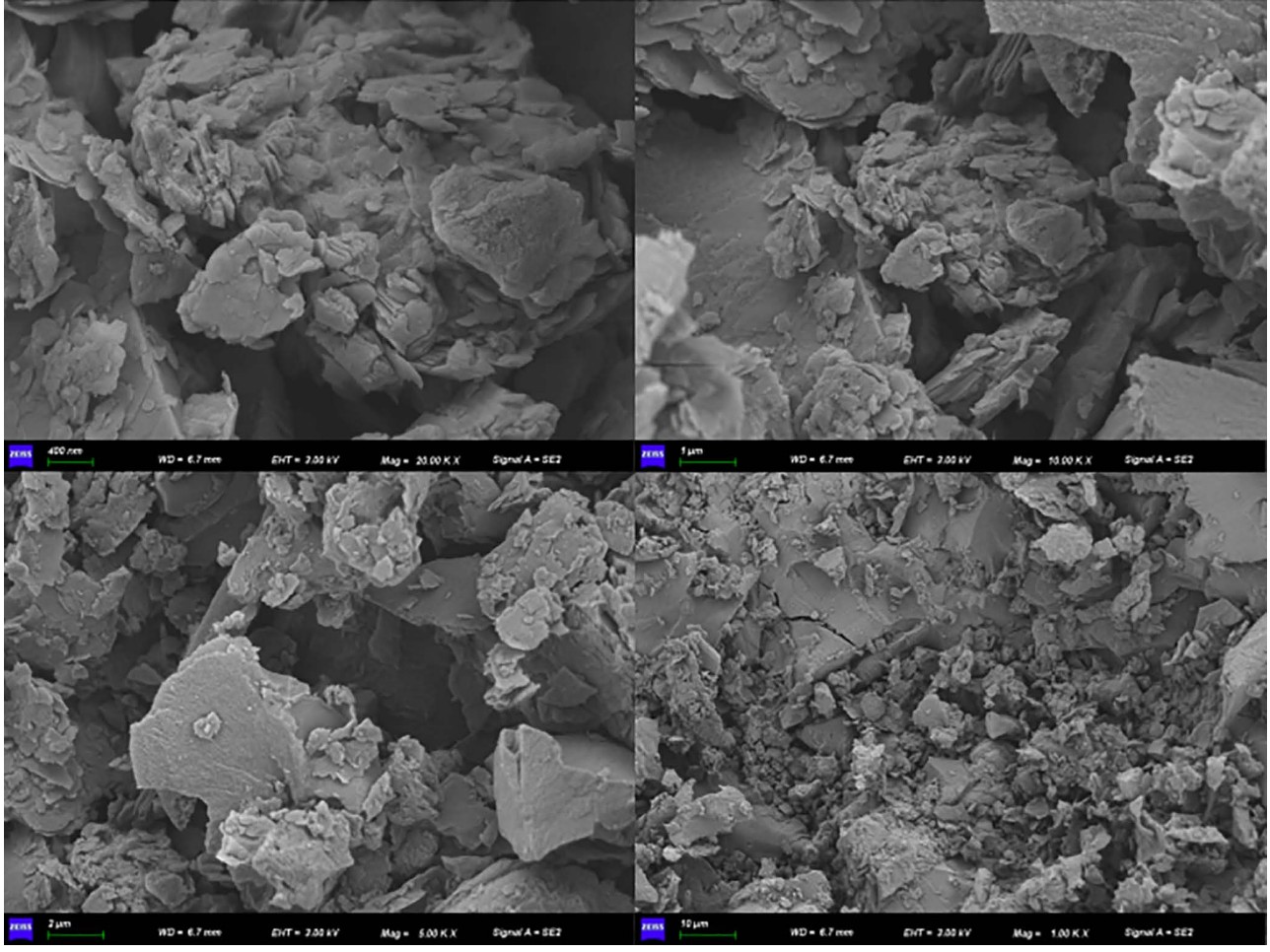

**Fig 1. SEM test results.**

## 2.3 Low-temperature N$_2$ adsorption test

The low-temperature N$_2$ adsorption-desorption curve of the coal sample is shown in Fig 2. According to the IUPAC classification standard, this curve is a combination of type II and type IV(a) curves. In the low-pressure section, the slope of the adsorption curve is relatively large. At this time, the main adsorption mechanism is micro-pore filling. It can be observed that the adsorption and desorption curves do not overlap, and there is a desorption lag phenomenon. The lag line belongs to type H4, indicating that there are a certain number of ink bottle pores.

## 2.4 Experimental apparatus and procedures

As shown in Fig 3, the gas adsorption experimental device's principle was illustrated. The main experimental steps were as follows:

1. A certain mass of coal sample was packed into the coal sample can, with an appropriate amount of swabs applied to the top, and the water bath was adjusted to the specified temperature.

2. The vacuum pump was turned on to evacuate the experimental system.

3. The reference gas was charged into the reference vessel at a pressure of 1–5 MPa. The valve between the reference and coal-sample vessels was then opened to initiate the gas-adsorption process.

4. After adsorption equilibrium was achieved, step 3 was repeated for the next pressure – point adsorption.

5. Throughout the experiment, the pressure sensor recorded pressure data in real time and transmitted it to the computer.

## 3 Analysis of experimental results

### 3.1 Adsorption process

As shown in Fig 4, the gas adsorption – diffusion process was depicted. From 0 to 12 min, the volume of adsorbed gas rose sharply as the gas flowed into the coal sample cell. Over time, the adsorption per unit time gradually decreased, the adsorption curve became more gentle and eventually equilibrium was reached. Comparing the gas volume adsorbed at

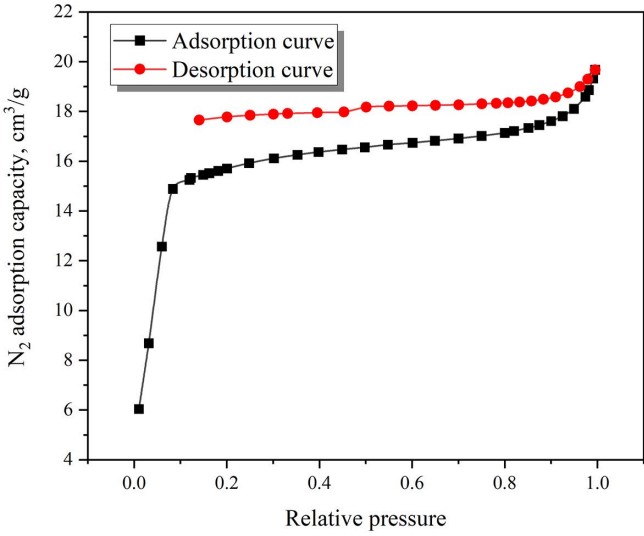

**Fig 2. Low temperature N$_2$ adsorption test results.**

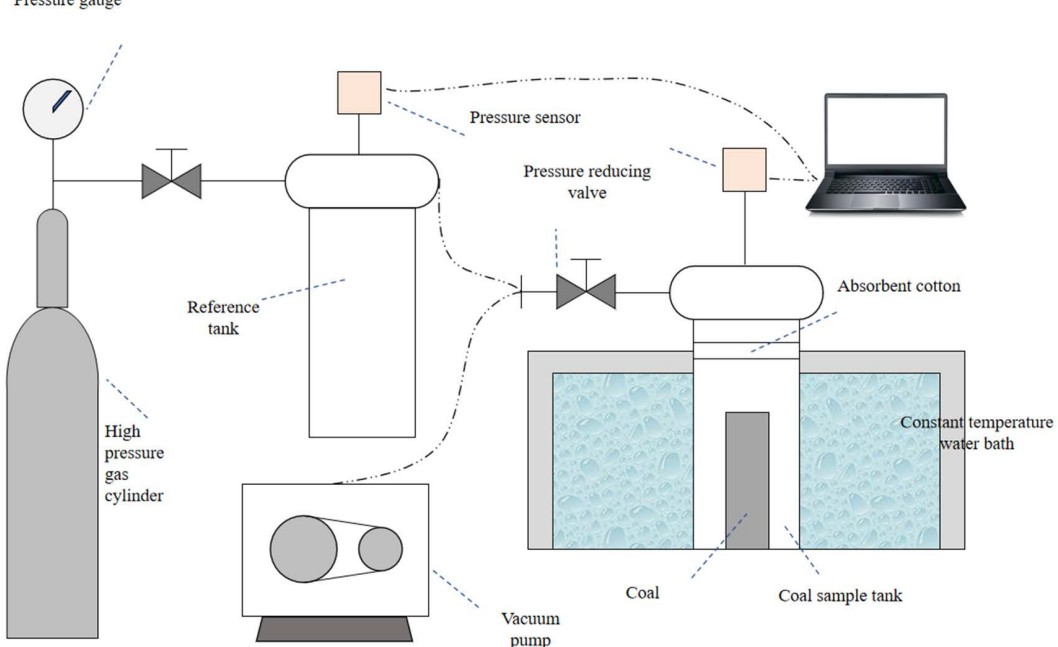

**Fig 3. Schematic diagram of experimental setup.**

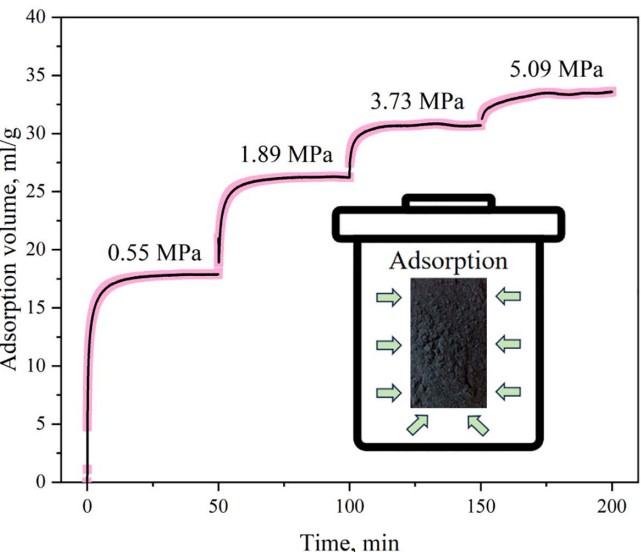

**Fig 4. Gas adsorption process curve.**

different equilibrium pressures, it was found that as equilibrium pressure increased, the incremental adsorbed gas volume gradually reduced, indicating a finite adsorption capacity as pressure approached infinity. On the one hand, adsorption occurred under constant – volume conditions, causing the gas pressure in the coal sample can to decline gradually. On the other hand, as adsorption proceeded, gas molecules migrated deeper into coal particles, lengthening the migration

 

path. Moreover, adsorption sites in coal reduced continuously during adsorption. These three factors jointly shaped the characteristics of the gas adsorption curve.

Theories of gas adsorption included the monolayer theory, multilayer theory, and pore – filling theory, which corresponded to the Langmuir equation, BET equation, and DR equation respectively [18–20].

$$Q = \frac{abp}{1 + bp}$$

(1)

In the equation, $a$ and $b$ were the Langmuir adsorption constants, $p$ was the gas pressure in MPa, and $Q$ was the adsorption capacity in milliliters per gram.

$$n_a = \frac{n_m c p}{(p_0 - p)\left[1 + (c - 1)(p/p_0)\right]}$$

(2)

In the equation, $n_a$ was the methane adsorption capacity, mol/kg; $n_m$ was the monolayer adsorption limit, mol/kg; $c$ was constant linked to adsorption heat; $p_0$ was the saturated vapor pressure of the adsorbed gas.

$$V_m = V_{mic} \exp\left\{-\left[\frac{RT}{\beta_0 E_0} \ln\left(\frac{p_0}{p}\right)\right]^2\right\}$$

(3)

In the formula, $V_m$ represents the adsorption capacity of gas, m³/kg; $V_{mic}$ is the volume of microporous structure, m³/kg; $R$ is the general gas constant, J/(molK); $T$ is the temperature, K; $\beta_0$ is the affinity coefficient of the adsorbed molecule, with no dimensionality. $E_0$ is the characteristic adsorption energy, J/mol.

The above three equations were used to fit the gas adsorption volume, as shown in Fig 5. It can be seen that the DR equation and the Langmuir equation had a good fit, with R2 values of 0.9986 and 0.9976 respectively. In comparison, the BET equation had a poorer fit, but its fit was remained relatively high at 0.9765. According to previous research results, the Langmuir equation has a good fit for most coals, and its form is simple with clear parameter meanings, making it the most widely used adsorption model at present. The micropores in coal are well developed, and gas molecules stored in the form of micropore filling account for a large proportion, which to some extent promoted the good fit of the DR equation.

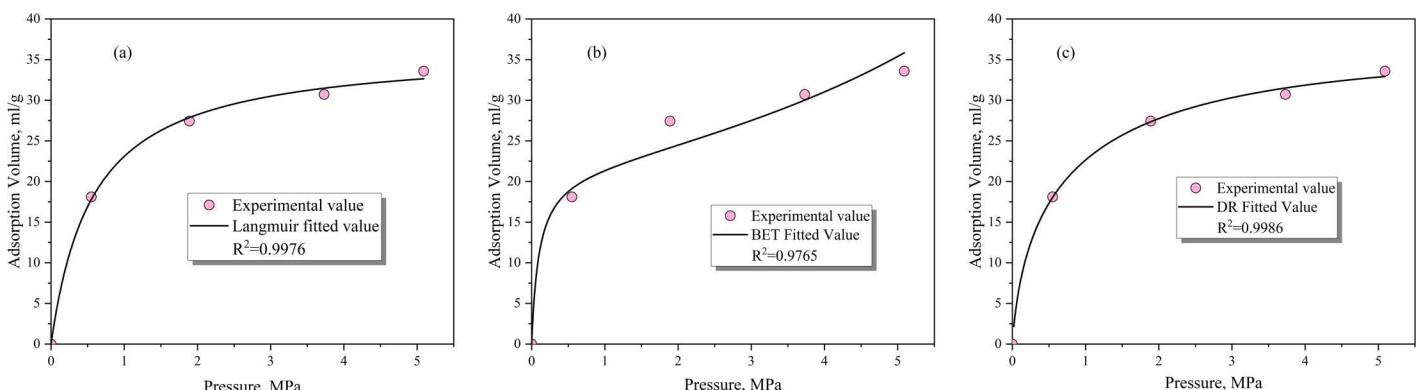

**Fig 5. Fitting effect of gas adsorption model.**

## 3.2 Desorption process

At a pressure of 1 MPa and a temperature of 40 °C, we measured the final desorption volume of the coal sample to be 17.7 cm³/g. This value is regarded as the theoretical limit under the constant pressure (atmospheric pressure) boundary condition; however, as shown in Fig 6, due to factors such as diffusion retardation and matrix contraction, the measured desorption volume only reached 14.7 cm³/g, which is equivalent to 83.1% of the theoretical value. This result is consistent with that reported in Reference [21] under similar conditions (0.8 MPa, 30 °C), and it also conforms to the increasing trend with the increase in initial pressure, thereby indirectly verifying the accuracy of this experiment. Analysis revealed two main reasons for this discrepancy. Firstly, some gas leaks can cause the actual gas pressure or concentration to deviate from the set value, thereby underestimating the adsorption capacity by 1% − 4%, the permeability by 5% − 7%, and the deformation error of the coal body by approximately 0.05 - 0.2 × 10⁻² mm; the leakage effect is more pronounced at high temperatures. Secondly, gas adsorption induced swelling and deformation of the coal matrix. After such swelling [22,23], the size of some molecular-scale pore throats became smaller than gas molecules, creating a blocking effect on gas molecules within the pores.

# 4 Simulation of diffusion process

## 4.1 Model

In this paper, the most widely used single-pore model, double-hole model and dynamic diffusion coefficient model at present are adopted to simulate the gas diffusion process [24–26], and the applicability of the model and the diffusion mechanism are studied.

The single-pore model can be expressed as:

$$\left[\rho_c \frac{p_{st}M}{RT_{st}} \frac{V_L P_L}{(P_L + p)^2} + \frac{\varphi M}{RT}\right]\frac{\partial p}{\partial t} - D\frac{M}{RT}\nabla p = 0$$

(4)

In the formula, $\rho_c$ represents the coal density; $\varphi$ for porosity; $D$ is the diffusion coefficient and t is the time.

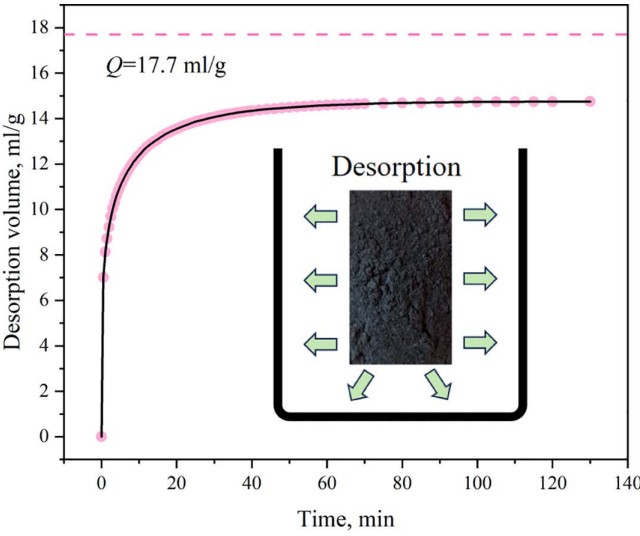

**Fig 6. Gas desorption process curve.**

The two-hole model uses two equations of large-hole diffusion and small-hole diffusion, which can be expressed as:

$$\left[\beta_d \rho_c \frac{p_{st}M}{RT_{st}} \frac{V_L P_L}{(P_L + p_d)^2} + \beta_d \frac{\varphi M}{RT}\right] \frac{\partial p_d}{\partial t} - D_d \frac{M}{RT} \nabla p_d = 0 \tag{5}$$

$$\left[\beta_x \rho_c \frac{p_{st}M}{RT_{st}} \frac{V_L P_L}{(P_L + p_x)^2} + \beta_x \frac{\varphi M}{RT}\right] \frac{\partial p_x}{\partial t} - D_x \frac{M}{RT} \nabla p_x = 0 \tag{6}$$

In the formula, $\beta_d$ and $\beta_x$ are the adsorption proportion coefficients of macropores and micropores respectively; The subscripts $d$ and $x$ represent large holes and small holes respectively.

The dynamic diffusion coefficient model regards the diffusion coefficient as a quantity that varies over time and can be expressed as:

$$\left[\rho_c \frac{p_{st}M}{RT_{st}} \frac{V_L P_L}{(P_L + p)^2} + \frac{\varphi M}{RT}\right] \frac{\partial p}{\partial t} - D_0 \exp(-\beta t) \frac{M}{RT} \nabla p = 0 \tag{7}$$

## 4.2 Model applicability

The results of the gas adsorption and desorption process simulation are presented in Figs 7 and 8, with the key model parameters listed in Table 1. For the adsorption process, all three models demonstrated a good fit, among which the dual-pore model had the best fitting effect, followed by the dynamic diffusion coefficient model, and the single-pore model had a relatively poor fit. However, due to its simplicity and the fact that its main parameter is a unique and constant diffusion coefficient, the single-pore model is suitable for simulating the gas adsorption process. Unlike the adsorption process, the single-pore model performed poorly in simulating the gas desorption process, as it underestimated the gas desorption

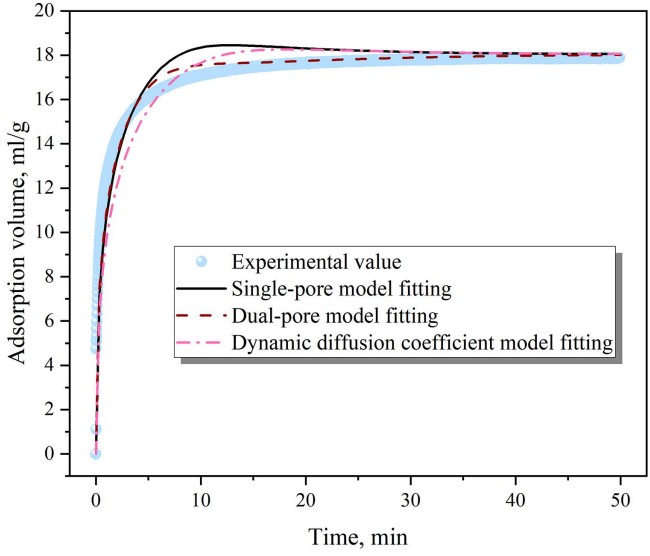

**Fig 7. Fitting of gas adsorption process.**

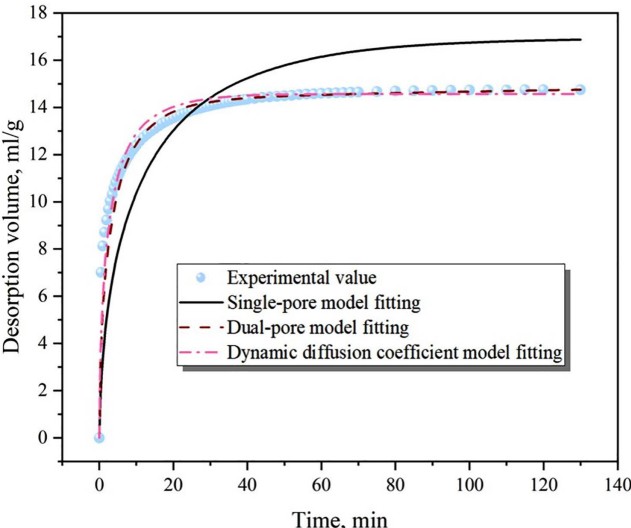

**Fig 8. Fitting of gas desorption process.**

**Table 1. Model key parameters.**

| Parameters | Value | |
|---|---|---|
| | **Adsorption** | **Desorption** |
| $a$ | 36.32 m³/t | 36.32 m³/t |
| $b$ | 1.74 MPa | 1.74 MPa |
| $\varphi$ | 2.4% | 2.4% |
| $M$ | 0.016 kg/mol | 0.016 kg/mol |
| $R$ | 8.314 J/(mol·K) | 8.314 J/(mol·K) |
| $T$ | 313.15 K | 313.15 K |
| $D$ | $1.9 \times 10^{-11}$ m²/s | $5 \times 10^{-12}$ m²/s |
| $D_d$ | $2.1 \times 10^{-11}$ m²/s | $1.4 \times 10^{-11}$ m²/s |
| $D_x$ | $1 \times 10^{-12}$ m²/s | $7 \times 10^{-15}$ m²/s |
| $\beta_d$ | 0.8 | 0.835 |
| $D_0$ | $1.5 \times 10^{-11}$ m²/s | $1.5 \times 10^{-11}$ m²/s |
| $\beta$ | $1 \times 10^{-4}$ s⁻¹ | $1.6 \times 10^{-3}$ s⁻¹ |

amount in the early diffusion stage and overestimated it in the later stage. In contrast, both the dual-pore model and the dynamic diffusion coefficient model showed good fitting accuracy and are more appropriate for simulating the gas desorption process.

The applicability of the three models for coal-seam gas migration was further contemplated. On the one hand, during gas drainage, the diverse pore structures in the coal matrix led to varying migration resistances for gas molecules in different pores, causing the diffusion coefficient to change over time. On the other hand, at the same time, the gas pressure and diffusion stages varied at different locations in the coal seam, creating spatial differences in the diffusion coefficient as well. The single-pore model, with a diffusion coefficient that remains constant in both time and space, struggled to accurately predict the characteristics of gas migration in coal seams. The dynamic diffusion coefficient model allowed for

variations in the diffusion coefficient over time and matched well with coal particle gas diffusion experiments. However, its diffusion coefficient was assumed to be a constant value in space, equivalent to a spatially averaged value. For gas migration at the coal-seam scale, if the gas pressure at a certain point remained unchanged (i.e., the drainage process did not affect that area), the diffusion coefficient still decayed similarly to that at the borehole location in this model, which was inconsistent with actual conditions. In contrast, the dual-pore model, although having two constant diffusion coefficients, allowed for different weights of macropore and micropore diffusion at different locations. As a result, its overall diffusion coefficient exhibited spatial evolution characteristics. Therefore, the dual-pore model was found to be highly applicable for simulating gas migration at the coal-seam scale.

### 4.3 Gas pressure distribution

In order to facilitate analysis, monitoring points M1(0, 0, 0), M2(0.05, 0, 0), and M3(0.1, 0, 0) were set up. Taking the dual – pore model as an example, the pressure distribution law during gas diffusion was analysed. Considering the weights of gas pressure in macropores and micropores, gas pressure was defined as follows:

$$p = \beta_d p_d + \beta_x p_x \tag{8}$$

As shown in Fig 9, during adsorption, gas pressure propagated from the external areas of the coal particle toward the center, while during desorption, the pressure moved in the opposite direction, from the center to the exterior. Though seemingly reversible, these processes had distinct differences. Adsorption occurred under constant volume conditions, with the volume of free gas fixed and pressure decreasing over time. Desorption, however, took place under constant pressure conditions, with the pressure of free gas held constant and its volume increasing as desorption proceeded. Within the experimental timeframe, adsorption led to gas pressure homogenization between the coal particle's interior and the external environment, with pressure equilibrium approached throughout the particle. In contrast, during desorption, a persistent pressure gap remained between the internal areas of the coal particle and the external pressure, with a gas pressure gradient sustained within the particle.

## 5 Conclusion

In this study, experiments on gas adsorption-desorption for particulate coal were conducted, followed by numerical simulations based on the experimental findings, to explore gas diffusion in coal. The key conclusions are as follows:

(1) By comparing the three classical equations (DR, Langmuir, and BET) with the adsorption system dominated by micropores in coal particles, it was found that the DR equation (with $R^2 = 0.9986$) significantly outperformed the Langmuir and BET equations, providing a new criterion for the micropore filling mechanism and breaking through the previous empirical limitation of only using the Langmuir equation.

(2) For the gas adsorption process, the single-pore, dual-pore, and dynamic diffusion coefficient models all showed good fitting results. However, due to its simplicity, the single-pore model is more applicable. For the gas desorption process, the single-pore model performed poorly, while the dual-pore and dynamic diffusion coefficient models showed good fitting results. When considering gas migration at the coal seam scale and the evolution of the diffusion coefficient in spatial and temporal dimensions, the dual-pore model demonstrates better applicability.

(3) The experimental-simulation coupling reveals the "adsorption-expansion-sealing" chain mechanism: It quantitatively determines the critical size for pore throat sealing (<0.38 nm), explains the "desorption lag" phenomenon commonly observed in the field, and fills the gap in the mechanism between micropore structure and macroscopic production capacity.

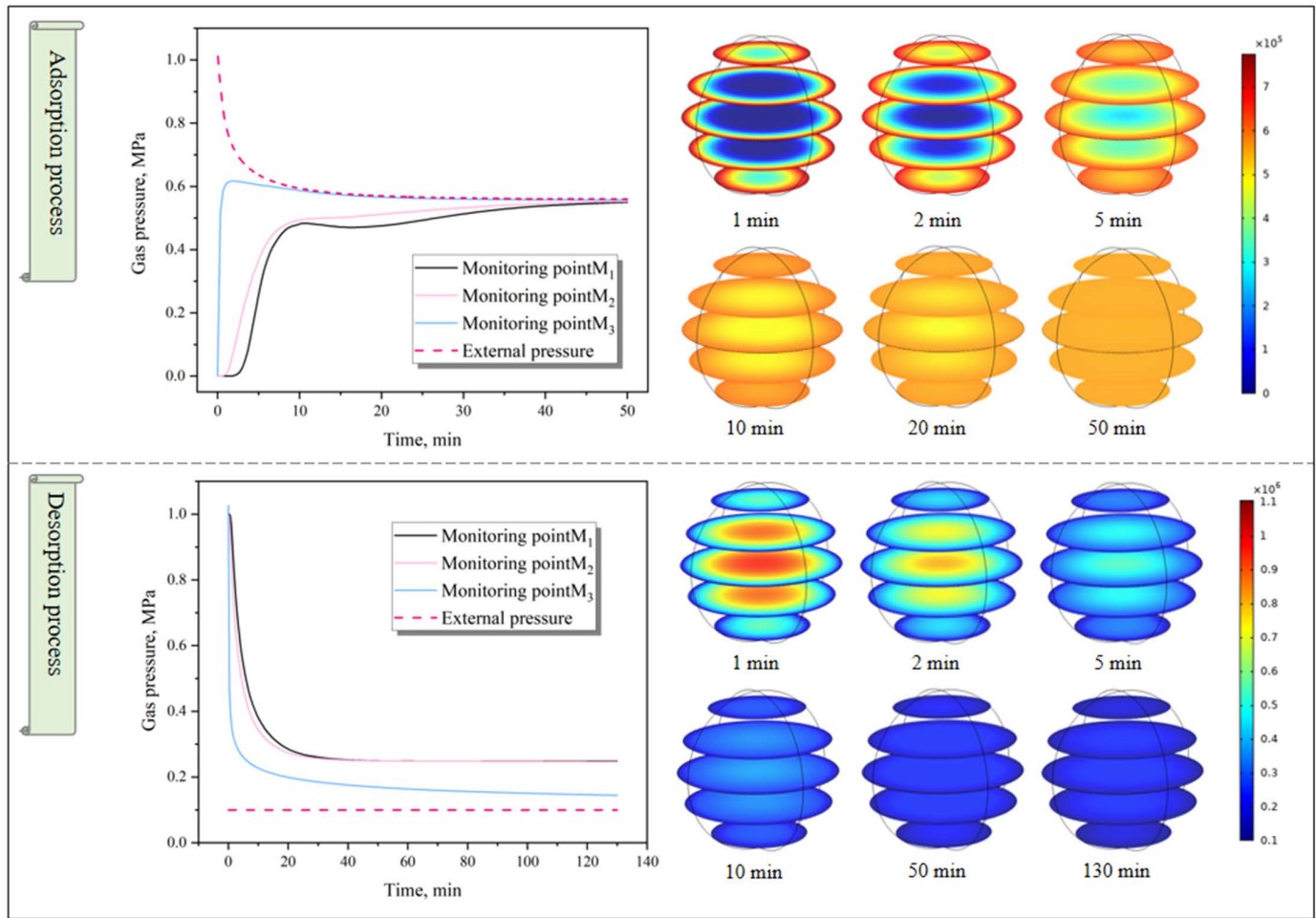

**Fig 9. Distribution characteristics of gas pressure.**

(4) There are specific differences between the adsorption process and the desorption process. The adsorption process is under constant volume conditions, while the desorption process is under constant pressure conditions. During the adsorption process, the gas pressure eventually becomes similar to the external pressure, and the gas pressure throughout the interior of the coal particles also tends to reach equilibrium. However, for the desorption process, there is always a certain gap between the gas pressure and the external pressure, and there is a certain gas pressure gradient within the coal particles, providing a theoretical basis for the on-site extraction system (pressure gradient vs continuous negative pressure).

## Author contributions

**Conceptualization:** Enyu Xu, Xijian Li.

**Data curation:** Ting Xia, Enyu Xu, Xijian Li, Shoukun Chen.

**Investigation:** Shoukun Chen.

**Methodology:** Shoukun Chen.

**Supervision:** Xijian Li.

**Validation:** Xijian Li.

**Writing – original draft:** Enyu Xu.

**Writing – review & editing:** Ting Xia, Enyu Xu.

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
