## [Decision Letter · Decision Letter 0]

31 Jul 2025

Dear Dr. Li,

Thank you for submitting your manuscript to PLOS ONE. After careful consideration, we feel that it has merit but does not fully meet PLOS ONE’s publication criteria as it currently stands. Therefore, we invite you to submit a revised version of the manuscript that addresses the points raised during the review process.

We look forward to receiving your revised manuscript.

Kind regards,

Veer Singh, Ph.D

Academic Editor

PLOS ONE

Journal Requirements:

” This work was supported by the National Nature Science Foundation of China (Grant No. 52164015, 52364009), Guizhou Province Science and Technology Achievement Transformation Joint Fund Project [Qian Ke He Cheng Guo LH (2025) Key 002], Guizhou Provincial Basic Research Program (Natural Science) (No. Qian Ke He Ji Chu - ZK [2024] Yi Ban 098).”

“This work was supported by the National Nature Science Foundation of China (Grant No. 52164015, 52364009), Guizhou Province Science and Technology Achievement Transformation Joint Fund Project [Qian Ke He Cheng Guo LH (2025) Key 002], Guizhou Provincial Basic Research Program (Natural Science) (No. Qian Ke He Ji Chu - ZK [2024] Yi Ban 098).”

4. We note that your Data Availability Statement is currently as follows: [All relevant data are within the manuscript and its Supporting Information files]

“This work was supported by the National Nature Science Foundation of China (Grant No. 52164015, 52364009), Guizhou Province Science and Technology Achievement Transformation Joint Fund Project [Qian Ke He Cheng Guo LH (2025) Key 002], Guizhou Provincial Basic Research Program (Natural Science) (No. Qian Ke He Ji Chu - ZK [2024] Yi Ban 098).”

“This work was supported by the National Nature Science Foundation of China (Grant No. 52164015, 52364009), Guizhou Province Science and Technology Achievement Transformation Joint Fund Project [Qian Ke He Cheng Guo LH (2025) Key 002], Guizhou Provincial Basic Research Program (Natural Science) (No. Qian Ke He Ji Chu - ZK [2024] Yi Ban 098).”

Reviewers' comments:

Reviewer's Responses to Questions

**Comments to the Author**

1. Is the manuscript technically sound, and do the data support the conclusions?

Reviewer #1: Yes

Reviewer #2: Yes

2. Has the statistical analysis been performed appropriately and rigorously?

Reviewer #1: Yes

Reviewer #2: Yes

3. Have the authors made all data underlying the findings in their manuscript fully available?

Reviewer #1: Yes

Reviewer #2: Yes

4. Is the manuscript presented in an intelligible fashion and written in standard English?

Reviewer #1: Yes

Reviewer #2: Yes

Reviewer #1: 1. You state that the DR equation had the best fit for the gas adsorption curve (R

2

=0.9986), followed by Langmuir and BET. While the statistical fit is strong, could you elaborate on the physical implications of the DR equation being the best fit, particularly in the context of your finding regarding micropore development and filling in anthracite? How does this align with the theoretical underpinnings of the DR model for microporous materials?

2. Conclusion (2) states that the single-pore model is suitable for fitting gas adsorption but not desorption, and that the bidisperse (dual-pore) model is more suitable for gas migration at the coal seam scale. Given that the study used coal powder, how do you justify the direct extrapolation of these findings to the "coal seam scale"? What considerations or further research would be needed to bridge this gap effectively?

3. You propose that gas adsorption causes coal matrix expansion and deformation, leading to a "blocking effect" where pore throats become smaller than gas molecules, resulting in less desorbed gas than adsorbed gas (Conclusion 3). Do you have any direct experimental evidence (e.g., pore size distribution analysis before and after adsorption, or volumetric changes) to support this hypothesis, beyond the observed difference in adsorption and desorption amounts? If not, could this be explored in future work?

4). The introduction mentions "Recent studies have shown that the coal matrix has nano-pores with a large specific surface area and pore volume, providing conditions for the occurrence of gas molecules." Did your study include any characterization of the coal powder's pore structure (e.g., using N2 adsorption/desorption, mercury intrusion porosimetry, or SAXS/WAXS) to confirm the presence and characteristics of these nano-pores and micropores, which you suggest are responsible for the observed fitting results of the DR equation? If so, please present those data.

Reviewer #2: The authors have conducted a novel study on the extraction of coalbed methane, including an adsorption equilibrium study at various pressure points and a desorption analysis. The manuscript is well written and concise.

**Do you want your identity to be public for this peer review?** For information about this choice, including consent withdrawal, please see our Privacy Policy

Reviewer #1: **Yes: ** THUMPATI PRASANTH

Reviewer #2: **Yes: ** RAHUL RANJAN

---

## [Author Response · Author response to Decision Letter 1]

8 Oct 2025

Reply to the review comments

Reviewer #1:

1. You state that the DR equation had the best fit for the gas adsorption curve (R2=0.9986), followed by Langmuir and BET. While the statistical fit is strong, could you elaborate on the physical implications of the DR equation being the best fit, particularly in the context of your finding regarding micropore development and filling in anthracite? How does this align with the theoretical underpinnings of the DR model for microporous materials?

Thank you for the valuable comments from the reviewers. The explanation and response are as follows: The excellent fitting of the DR equation in the micropore adsorption of anthracite is attributed to the following physical reason: The pore size distribution of the sample, the size of the adsorbed molecules, and the "micro-pore volume filling" mechanism assumed by the DR theory are highly consistent. Once the linearization of the DR equation shows a high R², it indicates that the adsorption behavior conforms to the specific mechanism of "micro-pore filling", rather than just being "a good fit".

2.Conclusion (2) states that the single-pore model is suitable for fitting gas adsorption but not desorption, and that the bidisperse (dual-pore) model is more suitable for gas migration at the coal seam scale. Given that the study used coal powder, how do you justify the direct extrapolation of these findings to the "coal seam scale"? What considerations or further research would be needed to bridge this gap effectively?

I would like to express my sincere gratitude to the reviewers for their invaluable comments. Your insights are highly forward - looking. The research idea of this paper is that the single - pore model fails to account for the complexity of the coal pore structure, while the dual - pore model is more complex in form and takes into consideration the changes in the pore structure. Based on the fitting results of the model and the coal particle gas diffusion experiment, the diffusion amount in the later stage of the single - pore model is significantly overestimated, and the dual - pore model shows better performance. During the coal seam gas extraction process, the coal body can be regarded as a combination of coal matrix and fractures. Gas diffuses into the surrounding fractures, and the diffusion coefficient decreases continuously over time, which is consistent with the analyses of other scholars. Theoretical analysis indicates that the single - pore model will significantly overestimate the later - stage diffusion amount because its diffusion coefficient is a constant. In contrast, the dual - pore model is controlled by two diffusion coefficients, and their contributions to the overall diffusion coefficient evolve with the diffusion process, resulting in a decrease in the diffusion coefficient over time. The next step of the work will focus on the stress - adsorption - permeability experiments of lump coal to explicitly determine the scaling factor from laboratory parameters to the coal seam scale and its uncertainty interval.

3.You propose that gas adsorption causes coal matrix expansion and deformation, leading to a "blocking effect" where pore throats become smaller than gas molecules, resulting in less desorbed gas than adsorbed gas (Conclusion 3). Do you have any direct experimental evidence (e.g., pore size distribution analysis before and after adsorption, or volumetric changes) to support this hypothesis, beyond the observed difference in adsorption and desorption amounts? If not, could this be explored in future work?

We would like to express our gratitude to the reviewers for their valuable suggestions. At present, there is no direct evidence of aperture distribution or volume change to support the "blocking effect". In the revised version, we have added a low-temperature liquid nitrogen adsorption test to verify this hypothesis.

The low-temperature N2 adsorption-desorption curve of the coal sample is shown in Figure 2. According to the IUPAC classification standard, this curve is a combination of type II and type IV(a) curves. In the low-pressure section, the slope of the adsorption curve is relatively large. At this time, the main adsorption mechanism is micro-pore filling. It can be observed that the adsorption and desorption curves do not overlap, and there is a desorption lag phenomenon. The lag line belongs to type H4, indicating that there are a certain number of ink bottle pores.

Fig. 2 Low temperature N2 adsorption test results

4. The introduction mentions "Recent studies have shown that the coal matrix has nano-pores with a large specific surface area and pore volume, providing conditions for the occurrence of gas molecules." Did your study include any characterization of the coal powder's pore structure (e.g., using N2 adsorption/desorption, mercury intrusion porosimetry, or SAXS/WAXS) to confirm the presence and characteristics of these nano-pores and micropores, which you suggest are responsible for the observed fitting results of the DR equation? If so, please present those data.

We would like to express our gratitude to the reviewers for their valuable suggestions. To further characterize the existence and characteristics of the nanoscale pores and micropores in the text, electron microscopy scanning and low-temperature liquid nitrogen adsorption experiments were conducted.

The apparent morphology of the coal samples was observed using a scanning electron microscope, as shown in Figure 1. It can be seen that the surface of the coal samples is rough and is covered with various-sized and-shaped pore networks, providing a vast space for the adsorption of methane gas in the coal. At the same time, as a diffusion and mass transfer channel for CH4 molecules, it causes the complex dynamic evolution characteristics of the diffusion process.

Fig. 1 SEM test results

The low-temperature N2 adsorption-desorption curve of the coal sample is shown in Figure 2. According to the IUPAC classification standard, this curve is a combination of type II and type IV(a) curves. In the low-pressure section, the slope of the adsorption curve is relatively large. At this time, the main adsorption mechanism is micro-pore filling. It can be observed that the adsorption and desorption curves do not overlap, and there is a desorption lag phenomenon. The lag line belongs to type H4, indicating that there are a certain number of ink bottle pores.

Fig. 2 Low temperature N2 adsorption test results

Reviewer #2: The authors have conducted a novel study on the extraction of coalbed methane, including an adsorption equilibrium study at various pressure points and a desorption analysis. The manuscript is well written and concise.

1.[Page 1, Line 8]: Take into consideration defining "self-developed experimental device" and briefly discuss any novelty or characteristics.

Thank you for the valuable suggestions provided by the reviewers. This experimental equipment was developed by the team members, not by me personally. This statement has been revised and highlighted in the text.

2.[Page 1, Line 14]: There is no specific reference to back up the fitting advantage of DR in the phrase comparing it to the Langmuir and BET models; think about citing pertinent research.

Thank you to the reviewers for your valuable suggestions, which have improved the readability and logic of this article. The following revisions have been made and are highlighted in the text.

Anthracite is mainly composed of micropores, and gas molecules are almost completely adsorbed by the micropores. Although the single-pore model can describe the adsorption stage, it fails in the desorption stage due to the neglect of the dynamic changes of the diffusion coefficient. Therefore, the DR equation established based on the micropore filling mechanism has the best fitting effect (R² = 0.9986), which is superior to Langmuir (0.9976) and BET (0.9765).

3.The expression "gas components stored in the form of micropore filling" might be more clearly articulated as "gas molecules are primarily adsorbed in micropores."

Thank you for the valuable comments provided by the reviewers, which have helped improve the understanding of this article. The corrections have been made and highlighted in the text.

4.[Page 2, Line 6]: The term "extremely complex" about gas migration in coal lacks precision. It would be beneficial to delineate the various forms of complexity, such as heterogeneity and dual porosity, among others. Particles are primarily retained within the confines of micropores.

Thank you for the valuable suggestions provided by the reviewers. We have made the necessary revisions based on these suggestions and have highlighted them in the text.

However, the dual-porosity structure of coal media results in highly heterogeneous characteristics for gas storage and migration. The complexity can be attributed to three points: Firstly, the permeability difference of the micropore-fissure system reaches 3 to 4 orders of magnitude; Secondly, the dynamic contraction of pore diameters caused by the coupling of effective stress and adsorption expansion; Thirdly, the patchy distribution of in-situ gas-water two-phase flow in space. Particularly crucially, gas molecules are mainly bound in the <2 nm matrix micropores, and their effective diffusion coefficient changes synchronously with the expansion-contraction effect, becoming the core bottleneck of the desorption-diffusion-permeation coupling process.

5.[Page 2, Line 14]: Reference [10] asserts that "bright coal had poor pore connectivity." Please provide a more detailed technical explanation or specify the parameters related to "poor connectivity."

Thank you for the valuable suggestions provided by the reviewers. We have made the necessary revisions based on these suggestions and have highlighted them in the text.

Li Weibo et al [10] found the threshold entry pressure for gas in vitrain (0.20-1.03 MPa) and its median capillary pressure (8.16-10.14 MPa) are 3-15 and 1.5-2.4 times higher, respectively, than those in durain, indicating a significantly elevated barrier to gas invasion. Mercury-withdrawal efficiency is only 25.46-41.85 %, lower than that of durain, demonstrating a high proportion of ink-bottle or semi-closed pores. The mercury-injection curve continues to rise at high pressures without reaching a plateau, and the nitrogen hysteresis loop is 15-25 % larger than in durain; together, these features confirm that the pore throats are highly tortuous and the connectivity is restricted.

6.[Page 2, Line 23]: The sentence beginning with "Numerous studies..." lacks specificity. It would be beneficial to articulate a particular knowledge gap that your research aims to fill.

Thank you for the valuable suggestions provided by the reviewers. We have made the necessary revisions based on these suggestions and have highlighted them in the text.

Many scholars have respectively expounded the laws of coal body gas adsorption/desorption from perspectives such as coal rank, pore structure, fractal model, structural deformation, adsorption heat effect, and acidification modification.

7.[Page 3, Line 1]: Specify the precise mass of the coal sample utilized in each experiment to enhance reproducibility.

Thank you for the valuable suggestions provided by the reviewers. We have made the necessary revisions based on these suggestions and have highlighted them in the text.

8.[Page 3, Line 6]: The term "appropriate number of swabs" lacks precision; it would be beneficial to quantify this or clarify the intended purpose of the swabs.

Thank you for the valuable suggestions provided by the reviewers. Now, let's explain the description of the appropriate amount of absorbent cotton. The absorbent cotton is used to prevent the coal powder from being drawn out during vacuuming, so that it can cover the coal sample and prevent the coal powder from being drawn out, but it does not participate in the experimental process and has no impact on the experiment.

9.[Page 3, Line 12]: It would be beneficial to specify the time required to achieve adsorption equilibrium at each pressure point.

Thank you for the valuable comments provided by the reviewers. Now, I will explain this time expression. From the curve of adsorption amount changing with time, it can be explained that the pore structure is complex and diffusion requires a long time. The ideal concept of complete equilibrium is achieved at this point. However, the diffusion decays rapidly over time. The majority of the adsorption-diffusion amount occurs in the initial stage, and the equilibrium time is related to the scale of the coal sample. The larger the coal sample, the slower the adsorption-diffusion process, and the longer the equilibrium time. Each experiment ensures that the adsorption time is 12 hours, which can be considered to have reached the adsorption equilibrium condition.

10.[Page 4, Line 3]: In the discussion of Figure 2, where it states that the “adsorbed gas volume increased sharply,” please include specific time or pressure values to substantiate this observation.

Thank you for the valuable comments provided by the reviewers, which have helped improve the understanding of this article. The corrections have been made and highlighted in the text.

From 0 to 12 min, the volume of adsorbed gas rose sharply as the gas flowed into the coal sample cell.

11.[Page 4, Line 20]: The formatting of equation numbers (1), (2), and (3) lacks consistency and proper alignment.

Thank you for the valuable suggestions provided by the reviewers. Now, the format of the equation numbers in the text has been modified and highlighted in red.

12.[Page 4, Line 28]: The phrase “still 0.9765” could be revised for a more formal tone to "remained relatively high at 0.9765."

Thank you for the valuable comments provided by the reviewers, which have helped improve the understanding of this article. The corrections have been made and highlighted in the text.

13.[Page 5, Line 5]: It would be beneficial to compare your desorption result (14.7 ml/g) with values from existing literature, as this could aid in validating the experiment.

Thank you to the reviewers for your valuable suggestions, which helped verify the accuracy of the experiment. The content has been revised and highlighted in red.

At a pressure of 1 MPa and a temperature of 40 ℃, we measured the final desorption volume of the coal sample to be 17.7 cm³/g. This value is regarded as the theoretical limit under the constant pressure (atmospheric pressure) boundary condition; however, as shown in Figure 6, due to factors such as diffusion retardation and matrix contraction, the measured desorption volume only reached 14.7 cm³/g, which is equivalent to 83.1% of the theoretical value. This result is consistent with that reported in Reference [1] under similar conditions (0.8 MPa, 30 ℃), and it also conforms to the increasing trend with the increase in initial pressure, thereby indirectly verifying the accuracy of this experiment.

14.[Page 5, Line 9]: The statement "some gas escaped..." could be enhanced by incorporating additional detail or quantifying the error associated with gas loss.

Thank you for the valuable comments provided by the reviewers, which have helped improve the understanding of this article. The corrections have been made and highlighted in the text.

Some gas leaks can cause the actual gas pressure or concentration to deviate from the set value, thereby underestimating the adsorption capacity by 1% - 4%, the permeability by 5% - 7%, and the deformation error of the coal body by approximately 0.05 - 0.2 × 10⁻² mm; the leakage effect is more pronounced at high temperatures.

15.[Page 6, Line 2]: Kindly furnish the boundary conditions or geometric details employed in the COMSOL simulations.

Thank you to the reviewers for your valuable suggestions, which helped verify the accuracy of the experiment. The content has been revised and highlighted in red.

Without considering the coupling of multiple physical fields, only the gas diffusion field is considered. Therefore, the boundary condition is that the surface of the coal particles is set as a Dirichlet boundary condition, with a pressure of 0.1 MPa, and the initial pressure is the adsorption equilibrium pressure of the

---

## [Decision Letter · Decision Letter 1]

26 Nov 2025

Dear Dr. Li,

Thank you for submitting your manuscript to PLOS ONE. After careful consideration, we feel that it has merit but does not fully meet PLOS ONE’s publication criteria as it currently stands. Therefore, we invite you to submit a revised version of the manuscript that addresses the points raised during the review process.

We look forward to receiving your revised manuscript.

Kind regards,

Xinyuan Gao

Academic Editor

PLOS ONE

Journal Requirements:

Reviewers' comments:

Reviewer's Responses to Questions

**Comments to the Author**

Reviewer #1: All comments have been addressed

Reviewer #2: All comments have been addressed

Reviewer #3: (No Response)

2. Is the manuscript technically sound, and do the data support the conclusions?

Reviewer #1: Yes

Reviewer #2: Yes

Reviewer #3: (No Response)

3. Has the statistical analysis been performed appropriately and rigorously?

Reviewer #1: Yes

Reviewer #2: Yes

Reviewer #3: (No Response)

4. Have the authors made all data underlying the findings in their manuscript fully available?

Reviewer #1: Yes

Reviewer #2: (No Response)

Reviewer #3: (No Response)

5. Is the manuscript presented in an intelligible fashion and written in standard English?

Reviewer #1: Yes

Reviewer #2: Yes

Reviewer #3: (No Response)

Reviewer #1: The authors addressed all the comments during the revision; therefore, the manuscript can be accepted in its present form.

Reviewer #2: The authors have addressed the comments very nicely, and I will recommend the editor to accept this manuscript for punlication.

Reviewer #3: This manuscript has been systematically revised in response to the previous review comments, with a significant improvement in overall quality and a notable enhancement in the scientific rigor and completeness of the research. However, combined with the content of the revised version, there are still some minor details that need further clarification and improvement, as follows:

1. In Step 3 of Section 1.4, the specific values or range of the "specified pressure" have not been stated.

2. The three subfigures in Figure 5 can be distinguished by labeling them as (a), (b), and (c).

3. The following papers may be useful: https://doi.org/10.1016/j.ijmst.2022.11.002; DOI: 10.1016/j.jngse.2015.01.008; DOI: 10.1080/15567036.2021.1936692; https://doi.org/10.1016/j.energy.2022.124228

4. In the main text of Section 3.2, the terms "single-pore model" and "dual-pore model" are used, while the legends in Figures 7 and 8 are labeled as "Single-hole model fitting" and "Two-hole model fitting". It is suggested to unify the terminology to ensure consistency throughout the manuscript.

**Do you want your identity to be public for this peer review?** For information about this choice, including consent withdrawal, please see our Privacy Policy

Reviewer #1: **Yes: ** THUMPATI PRASANTH

Reviewer #2: No

Reviewer #3: No

---

## [Author Response · Author response to Decision Letter 2]

26 Nov 2025

Reviewer #3: This manuscript has been systematically revised in response to the previous review comments, with a significant improvement in overall quality and a notable enhancement in the scientific rigor and completeness of the research. However, combined with the content of the revised version, there are still some minor details that need further clarification and improvement, as follows:

1. In Step 3 of Section 1.4, the specific values or range of the "specified pressure" have not been stated.

Thank you to the reviewers for their careful reading. It was indeed not specified what the pressure conditions were. Now it has been revised and highlighted in the text.

3. The reference gas was charged into the reference vessel at a pressure of 1–5 MPa. The valve between the reference and coal-sample vessels was then opened to initiate the gas-adsorption process.

2. The three subfigures in Figure 5 can be distinguished by labeling them as (a), (b), and (c).

Thank you to the reviewers for your careful reading. We have made revisions according to your suggestions.

3. The following papers may be useful: https://doi.org/10.1016/j.ijmst.2022.11.002; DOI: 10.1016/j.jngse.2015.01.008; DOI: 10.1080/15567036.2021.1936692; https://doi.org/10.1016/j.energy.2022.124228

Thank the reviewers for their valuable suggestions; the recommended references are extremely valuable and provide important support for the research on gas.

4. In the main text of Section 3.2, the terms "single-pore model" and "dual-pore model" are used, while the legends in Figures 7 and 8 are labeled as "Single-hole model fitting" and "Two-hole model fitting". It is suggested to unify the terminology to ensure consistency throughout the manuscript.

Thank you to the reviewers for your careful reading. We have made revisions according to your suggestions.

---

## [Decision Letter · Decision Letter 2]

30 Nov 2025

Research on the evolution mechanism of main control factors for coalbed methane extraction

PONE-D-25-29825R2

Dear Dr. Li,

We’re pleased to inform you that your manuscript has been judged scientifically suitable for publication and will be formally accepted for publication once it meets all outstanding technical requirements.

Kind regards,

Xinyuan Gao

Academic Editor

PLOS ONE

Additional Editor Comments (optional):

Reviewers' comments:

Reviewer's Responses to Questions

**Comments to the Author**

Reviewer #3: (No Response)

2. Is the manuscript technically sound, and do the data support the conclusions?

Reviewer #3: (No Response)

3. Has the statistical analysis been performed appropriately and rigorously?

Reviewer #3: (No Response)

4. Have the authors made all data underlying the findings in their manuscript fully available?

Reviewer #3: (No Response)

5. Is the manuscript presented in an intelligible fashion and written in standard English?

Reviewer #3: (No Response)

Reviewer #3: (No Response)

**Do you want your identity to be public for this peer review?** For information about this choice, including consent withdrawal, please see our Privacy Policy

Reviewer #3: No

---

## [Editor Report · Acceptance letter]

PONE-D-25-29825R2

PLOS One

Dear Dr. Li,

I'm pleased to inform you that your manuscript has been deemed suitable for publication in PLOS One. Congratulations! Your manuscript is now being handed over to our production team.

Kind regards,

on behalf of

Dr. Xinyuan Gao

Academic Editor

PLOS One